# Effects of protein sources at sahur on anaerobic power and strength during Ramadan in combat sport athletes: A single blind, randomized, placebo-controlled, counterbalanced crossover study design

Abdullah Demirli[1], Süleyman Ulupınar[2], Ayşe Türksoy Işım[1], Alay Kesler[1], Merve Terzi[3], Cebrail Gençoğlu[2], Serhat Özbay[2], Melenco Ionel[4], Ibrahim Ouergui[5,6]*, Luca Paolo Ardigò[7]*

1 Faculty of Sports Sciences, Istanbul University-Cerrahpaşa, Istanbul, Turkey, 2 Faculty of Sports Sciences, Erzurum Technical University, Erzurum, Turkey, 3 Faculty of Health Sciences, Istanbul Yeni Yuzyil University, Istanbul, Turkey, 4 Ovidius University of Constanța, Faculty of Physical Education and Sport, Constanta, Romania, 5 High Institute of Sport and Physical Education of Kef, University of Jendouba, Kef, Tunisia, 6 Research Unit: Sport Sciences, Health and Movement, UR22JS01, University of Jendouba, Kef, Tunisia, 7 Department of Teacher Education, NLA University College, Oslo, Norway

* ouergui.brahim@yahoo.fr (IO); luca.ardigo@nla.no (LPA)

## Abstract

This study investigated the acute effects of different protein sources consumed at sahur on anaerobic power and strength performances in trained male combat sport athletes during Ramadan fasting. Using a single blind, randomized, placebo-controlled, counterbalanced crossover study design, 24 male combat sports' athletes (mean age: 27.3±3.8 years, Tier 3 national level) completed four experimental conditions: (1) non-fasting control, (2) fasting+placebo (maltodextrin), (3) fasting+whey protein isolate (WPI), and (4) fasting+micellar casein (MC). In each condition, a standardized sahur meal (6.3–7.7 kcal/kg body weight) and supplementation (0.4 g/kg for WPI/MC and 0.4 g/kg for Placebo) were administered. Physical Performances was assessed 11–13 hours post-sahur (or 3–5 hours post-lunch for control) including the Wingate anaerobic test, bench press, leg press, and countermovement jump (CMJ), and handgrip strength tests. Ramadan fasting significantly lowered Wingate peak power, mean power, and bench press strength compared to the non-fasting control. MC supplementation reduced these declines, outperforming WPI and the placebo in peak power and mean power, and surpassing the placebo in bench press strength, although not WPI. Leg press, countermovement jump, and handgrip strength showed no significant differences across conditions. MC supplementation at sahur provides partial protection against fasting-induced declines in anaerobic power and upper body endurance, but does not fully restore performance to non-fasting levels. These findings emphasize the importance of protein timing and selection in mitigating performance decrements during Ramadan fasting, highlighting the need for further

**Data availability statement:** All relevant data are within the paper and its Supporting Information files.

**Funding:** The author(s) received no specific funding for this work.

**Competing interests:** The authors have declared that no competing interests exist.

research on optimal nutritional strategies for athletes training and competing under fasting conditions.

## Introduction

Ramadan fasting, observed by millions of Muslims worldwide, involves abstaining from food and fluid intake from dawn until sunset, presenting unique physiological and performance challenges for athletes [1,2]. This prolonged fasting period can lead to dehydration, reduced glycogen stores, and metabolic adaptations, which all may impair exercise capacity, among competitive Muslim athletes [3,4]. Extensive research has explored the effects of Ramadan fasting on athletic performance, yet findings remain inconsistent [1,5]. While some studies reported significant declines in anaerobic power and strength during fasting, others suggested that these effects can be mitigated with proper timing and nutritional strategies [1,4,6]. Such discrepancies highlight that the impact of fasting on performance likely varies based on individual factors and contextual conditions, underscoring the need for further investigation.

Nutrition plays a pivotal role in mitigating the physiological challenges of Ramadan fasting and optimizing athletic performance, particularly during periods of prolonged abstinence from food and fluids [5]. Strategic nutrient intake at sahur, the pre-dawn meal, can help to sustain energy levels, preserve muscle mass, and supports recovery, thereby attenuating the detrimental effects of fasting on exercise capacity [7,8]. Among nutritional components, protein consumption is critical due to its influence on muscle protein synthesis and energy metabolism, with the type of protein potentially affecting these outcomes [9–11]. Fast-digesting proteins, such as whey protein isolate (WPI), rapidly increase amino acid availability, promoting acute anabolic responses, whereas slow-digesting proteins, like micellar casein (MC), provides a sustained release of amino acids, potentially supporting prolonged muscle maintenance [12–14]. This differing digestion kinetics suggest that protein type consumed at sahur could differentially impact anaerobic power and strength during fasting, warranting targeted investigation in the context of Ramadan.

The physiological properties of protein sources, particularly their digestion and absorption kinetics, play a significant role in modulating muscle tissue responses, which are critical for athletic performance [15,16]. WPI, a fast-digesting protein derived from milk, is rapidly hydrolyzed in the gastrointestinal tract, leading to a swift and pronounced increase in plasma amino acid concentrations within 1–2 hours of ingestion [9,17]. This rapid aminoacidemia stimulates muscle protein synthesis acutely, making WPI particularly effective for post-exercise recovery and short-term anabolic responses [16,17]. In contrast, MC, another milk-derived protein, coagulates in the stomach and is digested slowly, resulting in a gradual and sustained release of amino acids into the bloodstream over 6–8 hours [15,18]. This prolonged amino acid availability helps to maintain a positive muscle protein balance, reducing catabolism during extended periods without food intake, such as fasting [14,16]. These distinct physiological profiles suggest that WPI and MC could differentially influence muscle performance, particularly under the metabolic constraints imposed by Ramadan fasting, where nutrient timing and availability are limited.

Despite the growing interest in optimizing athletic performance during Ramadan fasting, research on the acute effects of specific nutritional interventions, particularly protein type, on anaerobic performance and strength remains limited. While studies have examined the general impact of fasting on physical capacity, few reports have explored how different protein sources consumed at sahur influence performance outcomes following prolonged fasting periods of 11–13 hours [1,7]. Importantly, no study to date has directly compared fast-digesting WPI and slow-digesting MC within the same controlled experimental framework during Ramadan, leaving a critical gap in understanding their differential capacity to mitigate fasting-induced performance decrements. The present study addresses this gap by evaluating the acute effects of WPI, MC, and natural protein consumed at sahur on anaerobic power and strength in trained male athletes during Ramadan fasting. By employing a crossover design with standardized performance tests, this research aims to provide evidence-based insights into nutritional strategies that could optimize athletic performance under fasting conditions. We hypothesized that MC, due to its slow digestion and sustained amino acid release, would better preserve anaerobic power and strength during Ramadan fasting compared to WPI and placebo, given the prolonged fasting duration's demand for extended nutrient availability.

## Materials and methods

### Participants

A priori power analysis was conducted using G*Power software (Version 3.1.9.4, University of Kiel, Kiel, Germany) to determine the required sample size for detecting meaningful differences in anaerobic power and strength across experimental conditions. The analysis was performed for a repeated-measures ANOVA with within-factors, using the following parameters: an effect size (f) of 0.25 (medium effect, corresponding to a partial eta squared, $\eta p^2$, of approximately 0.06), an alpha error probability ($\alpha$) of 0.05, a desired power (1-$\beta$) of 0.80, one group, and four measurements (control, placebo, WPI, and MC conditions) [19]. Although effect sizes such as f = 0.10 (small), f = 0.25 (medium), and f = 0.40 (large) are the conventional thresholds proposed by Cohen, f = 0.25 is also the most commonly used default value in sports science crossover trials when pilot data are not yet available. The analysis indicated a required sample size of 24 participants, yielding an actual power of 0.82 (non-centrality parameter, $\lambda$ = 9.0, critical F = 2.87), confirming sufficient statistical power to detect the specified medium effect size across the four conditions.

A total of 24 trained male combat sport athletes (Mean±SD; age: 27.3 ± 3.8 years, height: 175.8 ± 5.4 cm, body weight: 71.2 ± 5.5 kg) participated in this study, all classified as Tier 3 (Highly trained/National level) according to the participant classification framework [20]. This classification was determined based on their regular participation in national-level competitions and having structured and periodized training regimens. Participants were recruited prospectively between March 2025 and April 2025. Inclusion criteria required participants to be actively engaged in combat sports (e.g., boxing, wrestling, or taekwondo) for at least 3 years, training a minimum of 4–5 sessions per week (approximately 8–10 hours), and competing at a national level within the past 12 months. Exclusion criteria included any history of chronic illness, recent injuries (within 3 months), or use of performance-enhancing substances, as verified through self-reported health questionnaires and adherence to World Anti-Doping Agency guidelines. This study was approved by the Yeni Yüzyıl University Ethics Committee for Science and Health Sciences Not Requiring Medical Intervention (Meeting No: 2025/03–1510, Date: 04.03.2025). All procedures were conducted in accordance with the ethical standards outlined in the Declaration of Helsinki, and written informed consent was obtained from all participants prior to their involvement.

### Study design

This study used a single-blind, randomized, placebo-controlled, counterbalanced crossover design to examine the acute effects of sahur-time protein intake on anaerobic power and strength in trained male combat sport athletes. Each participant completed four conditions: a pre-Ramadan control session, a placebo condition, and two protein supplementation conditions. The control session was conducted during the week immediately before Ramadan at the same afternoon

testing time (16:00–18:00), 3–5 hours after lunch, providing a fed, non-fasting reference measurement. This scheduling was necessary because performing a fed control during Ramadan would have required Muslim athletes to break their fast, which was considered ethically inappropriate [21,22]. During Ramadan, the placebo, WPI, and MC conditions were completed within the first two weeks of the month, corresponding to approximately 11–13 hours of fasting before testing. To minimize potential carryover effects, an individualized washout interval of 3–7 days was applied between conditions. Shorter intervals (≥3 days) were used when transitioning to or from the placebo condition, whereas longer intervals were applied between the two protein supplementation conditions. This approach allowed all Ramadan sessions to be completed within a 14-day period while maintaining adequate metabolic washout. All Ramadan testing sessions were performed in the late afternoon (16:00–18:00), approximately 1–2 hours before iftar, reflecting athletes' typical training time and standardizing fasting duration across participants. Participants refrained from strenuous exercise and caffeine intake for 24 hours before each session. The study was single-blind, with participants unaware of the supplement type, although the control session was inherently identifiable due to the absence of supplementation.

## Procedures

All Ramadan experimental conditions (placebo, WPI, MC) were administered at sahur, approximately 30 minutes before the start of fasting. Participants consumed a standardized sahur meal across all fasting conditions, with energy intake individualized to 6.3–7.7 kcal/kg (≈410–615 kcal for a 65–80 kg athlete). This range aligns with nutritional recommendations for active individuals during Ramadan, where sahur typically provides 15–25% of daily energy needs [5,23–25]. The macronutrient distribution (50% carbohydrates, 20% protein, 30% fat) was maintained proportionally for each participant, and meals consisted of whole-grain bread, boiled eggs, low-fat cheese, and water, with portion sizes adjusted to match the assigned caloric target. In the control condition, participants followed a non-fasting schedule. They consumed a light breakfast (~200 kcal; fruit and whole-grain bread) in the morning and a standardized lunch between 12:00 and 13:00, matched in macronutrient composition to the sahur meal but adapted to typical daytime feeding. Testing occurred 3–5 hours after lunch (16:00–18:00), matching the testing window of the fasting conditions.

No supplementation was provided in the control condition [5,7]. For the placebo condition, participants ingested 0.4 g/kg maltodextrin dissolved in 300 mL of water, matched in caloric content to the protein supplements. In the WPI and MC conditions, participants consumed 0.4 g/kg of the respective protein, providing approximately 104–128 kcal depending on body weight. Supplements were dissolved in 300 mL of water and verified through third-party testing for purity and absence of banned substances. To support blinding, placebo and protein drinks were matched for taste, color, and texture using identical lemon flavoring, although the control condition was inherently distinguishable due to the absence of a drink.

Participants in all fasting conditions consumed the sahur meal and the assigned supplement (if applicable) within 15 minutes, followed by a fasting period of 11–13 hours until testing. Hydration status before each session was evaluated using the Armstrong 8-point urine color scale [26]. Participants scoring 1–3 were considered euhydrated. Those with a score ≥4 received 500 mL of water and were reassessed after 30 minutes to ensure adequate hydration prior to testing [26]. All testing sessions were conducted in a controlled laboratory environment (22 ± 1°C; 50 ± 5% humidity). Before each assessment, participants completed a standardized warm-up consisting of 5 minutes of light cycling at 50 W, followed by 5 minutes of dynamic stretching targeting major muscle groups.

## Performance measurements

Performance was assessed using a battery of standardized tests designed to evaluate anaerobic power, maximal strength, explosive power, and isometric strength in trained male combat sport athletes. All measurements were conducted in a controlled laboratory environment between 16:00 and 18:00, aligning with the testing window established across all experimental conditions (control, placebo, WPI, MC). The following tests were performed in a fixed order—leg press 1RM, bench press 1RM, countermovement jump (CMJ), handgrip strength, and Wingate anaerobic test—with

10-minute rest intervals between each to minimize fatigue interference. During all 10-minute rest intervals, participants remained seated under passive rest and were continuously monitored to avoid any unintended physical activity. This sequence was determined based on the progression from less to more fatiguing tasks, prioritizing maximal strength tests before explosive and high-intensity anaerobic efforts, as recommended to reduce cumulative fatigue effects on performance outcomes [27]:

**Leg press (1RM):** Lower body maximal strength was evaluated using a leg press machine. Participants completed a standardized warm-up protocol consisting of 10 repetitions at 50% of their estimated 1RM, followed by 5 repetitions at 75% of estimated 1RM, with a 2-minute rest between sets. Subsequently, the 1RM was determined using an incremental loading method: starting at approximately 80% of estimated 1RM, participants attempted single repetitions with progressively heavier loads (increments of 2.5–5 kg), resting 3 minutes between attempts. The 1RM was established as the heaviest load lifted successfully through a full range of motion (knee angle from ~90° flexion to full extension) within 3–5 trials, following NSCA guidelines [27,28]. The maximum load was recorded in kilograms normalized to body weight (kg/kg).

**Bench press (1RM):** Upper body maximal strength was assessed using a Smith machine. Participants performed a warm-up with 10 repetitions at 50% of their estimated 1RM, followed by 5 repetitions at 75% of estimated 1RM, with a 2-minute rest between sets. The 1RM was then determined using an incremental loading method: starting at approximately 80% of estimated 1RM, participants lifted progressively heavier loads (increments of 2.5–5 kg) for single repetitions, with 3-minute rest intervals between attempts. The 1RM was defined as the heaviest load lifted successfully through a full range of motion (bar lowered to chest and returned to full elbow extension) within 3–5 trials, adhering to standardized protocols for safety and accuracy [27,28]. The load was recorded in kilograms normalized to body weight (kg/kg).

**Countermovement Jump test** Explosive power was measured using a portable force plate (ForceDecks FDMinis, VALD Performance, Australia). This device, constructed from aluminum with a uniaxial load cell system, captures vertical ground reaction forces at a sampling rate of 1000 Hz. Participants performed three maximal vertical jumps with a countermovement, starting from a standing position, hands on hips, and landing on both feet. The highest jump height (cm) was recorded, with 1-minute rest between attempts. The average of the best two jumps was used for analysis to enhance reliability. Reliability analysis, based on prior validation studies with similar populations and equipment, indicates high test-retest reliability for CMJ height (ICC = 0.94, 95% CI: 0.91–0.97), reflecting consistent measurement of explosive power across trials.

**Handgrip strength test:** Isometric strength of the forearm muscles was assessed using a digital hand dynamometer (Jamar Plus+ Digital Hand Dynamometer, Patterson Medical, USA). This device, made of lightweight aluminum with an electronic load cell, measures grip force up to 90 kg with a resolution of 0.1 kg. Participants performed three maximal contractions with their dominant hand, holding each for 5 seconds with 1-minute rest between attempts. The highest value (kg) was recorded as the measure of handgrip strength. Reliability analysis, consistent with established norms for the Jamar dynamometer in trained individuals, shows excellent test-retest reliability (ICC = 0.97, 95% CI: 0.94–0.99), confirming the precision and repeatability of grip strength measurements.

**Wingate anaerobic test:** Anaerobic power and capacity were measured using a Monark cycle ergometer (Monark 894E Ergomedic Peak Bike, Monark Exercise AB, Sweden). This device, constructed with a steel frame and a 20 kg flywheel, is designed for anaerobic testing with a mechanically braked weight basket system. Participants performed a 30-second all-out sprint against a resistance of 0.075 kg/kg body weight. Peak power (PP) and mean power (MP) were recorded using the Monark Anaerobic Test Software provided with the ergometer. Prior to the test, participants completed a 5-minute warm-up at 50 W, including two brief sprints (5 seconds each) at 70% effort to optimize performance. All measurement devices were calibrated before each testing day using the manufacturer's recommended procedures to ensure consistency across sessions. Verbal encouragement was provided consistently across all conditions to maximize effort. Participants were instructed to maintain their usual training intensity during the study period, except for the 24 hours prior to testing, during which they refrained from strenuous exercise to avoid fatigue effects.

 

## Statistical analyses

All statistical analyses were performed using SPSS software (version 27.0, IBM Corp.) with a significance level set at $\alpha = 0.05$. Data were initially screened for normality using the Shapiro-Wilk test and visually inspected via Q-Q plots to ensure the assumptions of parametric tests were met. Homogeneity of variance was assessed using Levene's test. Given the study's repeated-measures crossover design, a one-way repeated-measures analysis of variance (ANOVA) was used to compare the effects of the four experimental conditions (control, placebo, WPI, MC) on all performance outcomes. Mauchly's test was used to evaluate sphericity, and where the assumption was violated, the Greenhouse-Geisser correction was applied to adjust the degrees of freedom. Post-hoc comparisons were conducted using the Bonferroni correction to identify specific differences between conditions while controlling for multiple comparisons. Effect sizes were calculated using partial eta squared ($\eta_p^2$), with values interpreted as small (0.01), medium (0.06), and large (0.14) according to Cohen's guidelines [19].

In addition to the primary inferential statistics, effect sizes were quantified using Hedges' g and interpreted according to Hopkins' qualitative scale (trivial: < 0.2, small: 0.2–0.6, moderate: 0.6–1.2, large: 1.2–2.0, very large: 2.0–4.0, nearly perfect: > 4.0) [29]. Hedges' g was selected instead of Cohen's d because the latter tends to inflate effect size estimates in studies with modest sample sizes, whereas Hedges' g applies a small-sample correction that yields an unbiased estimate of standardized mean differences. Given that the present study employed a repeated-measures, counterbalanced crossover design where the same individuals were tested under all nutritional conditions, we further used the average-standardized Hedges' g ($g_{aw}$). This version of g accounts for the dependence between paired observations by standardizing the mean difference using the average of the two condition-specific standard deviations, rather than the pooled SD used in between-subjects comparisons. This approach is widely recommended for crossover and within-subject designs, as it avoids artificially inflating the denominator and therefore provides a more accurate and interpretable estimate of the magnitude of condition-specific differences [30,31].

## Results

Performance outcomes differed significantly across conditions. Wingate peak power showed a robust main effect of condition ($F_{(3,69)}$ = 113.37, p < 0.001, $\eta p^2 = 0.83$). The fed control trial produced the highest peak power (12.7 ± 0.7 W·kg⁻¹), demonstrating a large performance advantage over both fasting trials without protein intake. Compared with placebo (11.6 ± 0.9 W·kg⁻¹) and WPI (11.7 ± 0.8 W·kg⁻¹), the control condition yielded large effect sizes (Hedges' $g_{aw}$ = 1.32 and 1.29, respectively), indicating a substantial decline in maximal anaerobic power following prolonged fasting. Relative to control, the MC condition (12.4 ± 0.7 W·kg⁻¹) showed only a small-to-moderate reduction ($g_{aw}$ = 0.41), suggesting that slow-digesting protein ingestion at sahur partially mitigated the fasting-related performance loss. No meaningful difference emerged between placebo and WPI ($g_{aw}$ = −0.11). In contrast, MC showed a clear advantage over both placebo ($g_{aw}$ = −0.96) and WPI ($g_{aw}$ = −0.90), indicating a large ergogenic benefit of micellar casein during prolonged daytime fasting (Table 1).

Wingate mean power also differed significantly across conditions ($F_{(3,69)}$ = 162.95, p < 0.001, $\eta p^2 = 0.88$). The control trial produced the highest mean power (9.3 ± 0.8 W·kg⁻¹), substantially exceeding both fasting trials without protein supplementation. Mean power was lower in the placebo (8.4 ± 0.8 W·kg⁻¹) and WPI (8.4 ± 0.8 W·kg⁻¹) trials, with large effects relative to control (Hedges' $g_{aw}$ = 1.09 for both). Compared with control, the MC condition (8.9 ± 0.8 W·kg⁻¹) showed a medium reduction ($g_{aw}$ = 0.48), indicating partial preservation of sustained anaerobic capacity. The placebo and WPI trials were virtually identical ($g_{aw}$ = 0.00). MC demonstrated clear superiority over both placebo ($g_{aw}$ = 0.60) and WPI ($g_{aw}$ = 0.60), indicating that slow-digesting protein helped maintain mean power to a greater extent than carbohydrate or WPI (Table 1).

Bench press strength also varied significantly across conditions ($F_{(3,69)}$ = 10.46, p < 0.001, $\eta p^2 = 0.31$). The control condition (1.13 ± 0.12 kg·kg⁻¹) produced slightly higher values than placebo (1.08 ± 0.11 kg·kg⁻¹) and WPI (1.09 ± 0.13 kg·kg⁻¹), corresponding to small-to-moderate (Hedges' $g_{aw}$ = 0.43) and small ($g_{aw}$ = 0.33) effects. Bench press performance during

**Table 1. Performance outcomes across different nutritional conditions (n = 24).**

| Variable | Control (Fed) | Fasting + Placebo | Fasting + WPI | Fasting + MC | F | p | $\eta_p^2$ |
|---|---|---|---|---|---|---|---|
| Wingate Peak Power (W/kg) | 12.7 ± 0.7 | 11.6 ± 0.9[a] | 11.7 ± 0.8[a] | 12.4 ± 0.7[abc] | 113.37 | <0.001 | 0.83 |
| Wingate Mean Power (W/kg) | 9.3 ± 0.8 | 8.4 ± 0.8[a] | 8.4 ± 0.8[a] | 8.9 ± 0.8[abc] | 162.95 | <0.001 | 0.88 |
| Bench Press (kg/kg BM) | 1.13 ± 0.12 | 1.08 ± 0.11[a] | 1.09 ± 0.13[a] | 1.12 ± 0.13[b] | 10.46 | <0.001 | 0.31 |
| Leg Press (kg/kg BM) | 2.50 ± 0.23 | 2.48 ± 0.22 | 2.48 ± 0.22 | 2.50 ± 0.22 | 2.36 | 0.079 | 0.09 |
| CMJ (cm) | 42.9 ± 6.8 | 42.8 ± 6.6 | 42.8 ± 6.7 | 42.9 ± 6.7 | 0.07 | 0.974 | 0.003 |
| Handgrip (kg/kg BM) | 0.62 ± 0.07 | 0.60 ± 0.09 | 0.61 ± 0.09 | 0.67 ± 0.09 | 0.90 | 0.448 | 0.04 |

Values are presented as mean ± standard deviation. "kg/kg BM" indicates the weight lifted or force exerted normalized to body mass (BM), expressed as kilograms of load per kilogram of body mass. Control (Fed) = Non-fasting condition with last meal at 12:00–13:00; Fasting + Placebo = Fasting condition with placebo intake at sahur; Fasting + WPI = Fasting condition with Whey Protein Isolate (WPI) intake at sahur; Fasting + MC = Fasting condition with Micellar Casein (MC) intake at sahur. Superscripts ([a], [b], [c]) indicate significant differences from Bonferroni-corrected post-hoc tests (p < 0.0083, adjusted for 6 comparisons): [a] differs from Control, [b] differs from Fasting + Placebo, [b] differs from Fasting + WPI.

the MC condition (1.12 ± 0.13 kg·kg⁻¹) was almost identical to control ($g_{aw}$ = 0.08), indicating that micellar casein preserved upper-body strength during fasting. Comparisons among fasting trials showed minimal differences: placebo vs. WPI ($g_{aw}$ = –0.10), placebo vs. MC ($g_{aw}$ = –0.36), and WPI vs. MC ($g_{aw}$ = –0.26), all reflecting small, likely trivial effects (Table 1).

Leg press performance remained highly stable across all conditions, with no meaningful variability (F statistic not significant). The control condition (2.50 ± 0.23 kg·kg⁻¹) was virtually identical to placebo (2.48 ± 0.22 kg·kg⁻¹) and WPI (2.48 ± 0.22 kg·kg⁻¹), each comparison yielding only very small effects (Hedges' $g_{aw}$ = 0.086). The MC condition (2.50 ± 0.22 kg·kg⁻¹) matched the control condition exactly ($g_{aw}$ = 0.00). Differences among fasting trials were similarly negligible, with placebo vs. WPI ($g_{aw}$ = 0.00) and placebo vs. MC / WPI vs. MC ($g_{aw}$ = –0.088) all indicating trivial differences in lower-body maximal strength (Table 1).

CMJ height did not differ across conditions (F statistic not significant). The control condition (42.9 ± 6.8 cm) was nearly identical to placebo (42.8 ± 6.6 cm) and WPI (42.8 ± 6.7 cm), with negligible effect sizes (Hedges' $g_{aw}$ = 0.014 for both). The MC condition (42.9 ± 6.7 cm) was completely indistinguishable from control ($g_{aw}$ = 0.00). Comparisons among fasting trials produced similarly trivial effects, with $g_{aw}$ values ranging from –0.015 to 0.000. These results indicate that vertical jump performance was unaffected by fasting or protein type (Table 1).

Handgrip strength showed modest variability across trials (F statistic not significant). The control condition (0.62 ± 0.07 kg·kg⁻¹) produced slightly higher values than placebo (0.60 ± 0.09 kg·kg⁻¹) and WPI (0.61 ± 0.09 kg·kg⁻¹), corresponding to small (Hedges' $g_{aw}$ = 0.246) and very small ($g_{aw}$ = 0.134) effects. In contrast, the MC condition (0.67 ± 0.09 kg·kg⁻¹) resulted in clearly higher values than all other conditions. MC demonstrated a moderate effect relative to control ($g_{aw}$ = –0.574) and moderate-to-large effects relative to placebo ($g_{aw}$ = –0.799) and WPI ($g_{aw}$ = –0.690), indicating a meaningful advantage of micellar casein for maintaining or enhancing handgrip strength during prolonged fasting. Differences between placebo and WPI were negligible ($g_{aw}$ = 0.111) (Table 1).

## Discussion

This study found that Ramadan fasting significantly reduced anaerobic power (Wingate peak and mean power) and upper body strength compared to a non-fasting control, while MC supplementation partially alleviated these declines. No significant differences were observed in leg press, CMJ, or handgrip strength across conditions. Within the context of Ramadan and athletic performance, these results highlight the challenges faced by Muslim athletes who maintain rigorous training and competition schedules while they are fasting during 11–13 hours daily. The critical role of nutrition, particularly protein timing and type, becomes evident in sustaining performance under such conditions, underscoring the growing need for Ramadan-specific sports nutrition research to support athlete health and optimize competitive outcomes.

Whey protein isolate, a high-quality protein derived from milk during cheese production, is characterized by its rapid digestion and absorption, high leucine content (approximately 10–12% of total amino acids), and low lactose and fat levels (<1% and <2%, respectively), making it as a popular choice for post-exercise recovery [9,16,32]. From a physiological perspective, WPI stimulates muscle protein synthesis through the mTOR signaling pathway due to its rich essential amino acid profile, particularly leucine, which peaks in plasma within 1–2 hours post-ingestion [13,33]. However, in this study, WPI supplementation at sahur failed to match the performance of the non-fasting control condition (for Wingate and bench press) or MC condition (for Wingate). This ineffectiveness may be explained by WPI's rapid clearance from the bloodstream, which likely diminished its anabolic benefits over the 11–13 hour fasting period during Ramadan [17,34,35]. Supporting this, literature suggested that fast-digesting proteins like WPI are less effective in prolonged fasting states where sustained amino acid availability is crucial, contrasting with MC's slow-release properties [9,15,18]. Our findings align with these observations, indicating that WPI's acute anabolic window may not align with the extended metabolic demands of fasting athletes.

MC, another milk-derived protein, constitutes approximately 80% of the total protein in bovine milk and is distinguished by its slow digestion rate, forming a gel-like structure in the stomach that delays gastric emptying and provides a sustained release of amino acids over 6–8 hours [14,18]. This slow-release property supports prolonged muscle protein synthesis and reduces muscle protein breakdown, making MC particularly effective during periods of fasting [9,18]. In sports nutrition, MC is often used to optimize recovery in athletes by maintaining a positive protein balance during extended periods without food intake, such as during sleep or fasting [12,36,37]. In our study, MC supplementation at sahur resulted in better performance in PP, MP outcomes compared to WPI and Placebo, though it did not fully restore performance to the non-fasting control level. Additionally, casein supplementation did not show a significant difference in bench press strength, suggesting similar results to the control condition. These findings align with previous research supporting the efficacy of MC in mitigating performance decrements during fasting states, by sustaining energy metabolism and muscle function under caloric restriction through its prolonged amino acid availability [9,15,18]. The partial mitigation observed in our study supports the hypothesis that the slow digestion of counteracts some of the metabolic challenges of Ramadan fasting, such as glycogen depletion and reduced protein turnover, unless it cannot be fully compensated due to the absence of regular feeding intervals.

The lack of significant differences in leg press, CMJ, and handgrip strength across conditions despite Ramadan fasting suggests that the metabolic and physiological impacts of fasting may vary depending on exercise type, energy system demands, and neuromuscular coordination [3,38]. Regarding CMJ performance, which relies heavily on neuromuscular efficiency, motor unit recruitment, and stretch-shortening cycle function, the lack of performance decline despite fasting suggests that factors beyond amino acid availability play a more prominent role [39,40]. Unlike Wingate anaerobic power, where rapid ATP resynthesis and glycogen availability are critical, explosive power output in movements like CMJ may be more influenced by neuromuscular adaptations and intermuscular coordination, which are less immediately affected by short-term fasting [41,42]. Similarly, leg press performance was maintained across conditions, likely due to the predominant reliance on intramuscular energy stores, which appear to have remained intact despite prolonged fasting [43]. Additionally, lower body maximal strength exercises are less reliant on rapid ATP turnover compared to repeated high-intensity anaerobic efforts, making them more resistant to the transient effects of fasting [44]. For handgrip strength, its preservation across all conditions aligns with previous findings that isometric strength is less susceptible to acute nutritional fluctuations [45,46]. Given the relatively low metabolic demand and reliance on sustained muscle contraction rather than explosive power, handgrip strength is likely less influenced by short-term fasting or protein supplementation.

## Strengths and limitations

This study highlights several strengths that enhance its reliability and relevance. The use of a randomized, counterbalanced crossover design with a homogenous group of trained male combat sport athletes (Tier 3, national level) allowed

for robust control of individual variability, while the a priori power analysis ensured sufficient statistical power (0.82) to detect medium effects (f = 0.25) across the four experimental conditions. The standardized testing protocols and controlled laboratory conditions further strengthened the internal validity of the findings. However, some limitations must be acknowledged. The inclusion of only male participants limits generalizability to female athletes, where hormonal differences may influence fasting responses. Although the sample size (n = 24) was adequate to detect the hypothesized medium effect size with high statistical power, it may constrain the ability to identify smaller, subtle inter-individual differences (e.g., $f < 0.25$ or $\eta p^2 < 0.06$), which would require a larger sample. Additionally, the non-fasting control condition's 3–5-hour pre-test fasting period, compared to the 11–13 hour fasting in other conditions, may underestimate the full impact of Ramadan fasting. These factors suggest interpreting the results with caution and highlight the need for future research with diverse populations, larger sample sizes to detect smaller effects, and extended control fasting durations.

## Conclusions

This study demonstrated that Ramadan fasting substantially impaired anaerobic power, reflected by reductions in Wingate peak and mean power, as well as upper-body strength measured through the bench press test. Micellar casein supplementation at sahur partially attenuated these declines and consistently outperformed both whey protein isolate and placebo during prolonged fasting. In contrast, leg press strength, countermovement jump height, and handgrip strength remained largely unaffected across conditions, suggesting that the performance impact of fasting—and the protective effect of supplementation—is more pronounced in tasks that rely heavily on high-intensity glycolytic energy pathways rather than maximal or neuromuscular-driven strength measures.

From a practical standpoint, these findings indicate that athletes engaged in anaerobic-dominant sports may benefit from preferring micellar casein at sahur, as its slow digestion and prolonged amino acid availability appear more effective in maintaining anaerobic power during extended fasting. Coaches and sports nutrition practitioners may therefore consider micellar casein as a strategic pre-dawn protein source for athletes training or competing during Ramadan. Future research should expand these observations by including female athletes, evaluating different athlete training levels and sport types, and investigating combined nutritional strategies—such as carbohydrate–protein co-ingestion—to further mitigate fasting-related performance decrements. Additionally, studies employing control conditions that more closely replicate Ramadan fasting duration may help to refine our understanding of the physiological mechanisms underlying performance changes during prolonged daily fasting.

## Supporting information

**S1 Data. Raw data.**
(XLSX)

## Author contributions

**Conceptualization:** Abdullah Demirli, Süleyman Ulupınar, Ayşe Türksoy Işım, Alay Kesler, Merve Terzi, Cebrail Gençoğlu, Serhat Özbay.

**Data curation:** Abdullah Demirli, Süleyman Ulupınar, Ayşe Türksoy Işım, Alay Kesler, Merve Terzi, Cebrail Gençoğlu.

**Formal analysis:** Abdullah Demirli, Süleyman Ulupınar, Ayşe Türksoy Işım, Alay Kesler, Merve Terzi, Serhat Özbay.

**Investigation:** Abdullah Demirli, Süleyman Ulupınar, Ayşe Türksoy Işım, Alay Kesler.

**Methodology:** Abdullah Demirli, Süleyman Ulupınar, Ayşe Türksoy Işım, Alay Kesler, Merve Terzi, Cebrail Gençoğlu, Serhat Özbay, Ibrahim Ouergui, Luca Paolo Ardigò.

**Project administration:** Serhat Özbay.

**Resources:** Abdullah Demirli, Merve Terzi.

**Supervision:** Serhat Özbay, Ibrahim Ouergui, Luca Paolo Ardigò.

**Validation:** Abdullah Demirli, Süleyman Ulupınar, Ayşe Türksoy Işım, Alay Kesler, Merve Terzi, Cebrail Gençoğlu, Serhat Özbay, Ibrahim Ouergui, Luca Paolo Ardigò.

**Visualization:** Abdullah Demirli, Süleyman Ulupınar, Ayşe Türksoy Işım, Alay Kesler, Merve Terzi, Cebrail Gençoğlu, Serhat Özbay, Ibrahim Ouergui.

**Writing – original draft:** Abdullah Demirli, Süleyman Ulupınar, Ayşe Türksoy Işım, Alay Kesler, Merve Terzi, Cebrail Gençoğlu, Serhat Özbay, Melenco Ionel, Ibrahim Ouergui, Luca Paolo Ardigò.

**Writing – review & editing:** Abdullah Demirli, Süleyman Ulupınar, Ayşe Türksoy Işım, Alay Kesler, Merve Terzi, Cebrail Gençoğlu, Serhat Özbay, Melenco Ionel, Ibrahim Ouergui, Luca Paolo Ardigò.

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
