## [Decision Letter · Decision Letter 0]

5 Nov 2025

PONE-D-25-43284Effects of Protein Sources at Sahur on Anaerobic Power and Strength During Ramadan in Combat Sport Athletes: A Single Blind, Randomized, Placebo-Controlled, Counterbalanced Crossover Study DesignPLOS ONE

Dear Dr. Ouergui,

Thank you for submitting your manuscript to PLOS ONE. After careful consideration, we feel that it has merit but does not fully meet PLOS ONE’s publication criteria as it currently stands. Therefore, we invite you to submit a revised version of the manuscript that addresses the points raised during the review process.

**ACADEMIC EDITOR:** To move forward in the editorial process, a comprehensive revision responding to each reviewer’s comment is required.

We look forward to receiving your revised manuscript.

Kind regards,

Leonardo Vidal Andreato, PhD

Academic Editor

PLOS ONE

Journal Requirements:

3. In the online submission form, you indicated that data that support the findings of this study are available from the corresponding author upon reasonable request.

All PLOS journals now require all data underlying the findings described in their manuscript to be freely available to other researchers, either a. In a public repository, b. Within the manuscript itself, or c Uploaded as supplementary information.

Reviewers' comments:

Reviewer's Responses to Questions

**Comments to the Author**

1. Is the manuscript technically sound, and do the data support the conclusions?

Reviewer #1: Yes

Reviewer #2: Yes

Reviewer #3: Yes

2. Has the statistical analysis been performed appropriately and rigorously? 

Reviewer #1: No

Reviewer #2: Yes

Reviewer #3: Yes

3. Have the authors made all data underlying the findings in their manuscript fully available?

Reviewer #1: Yes

Reviewer #2: Yes

Reviewer #3: Yes

4. Is the manuscript presented in an intelligible fashion and written in standard English?

Reviewer #1: Yes

Reviewer #2: Yes

Reviewer #3: Yes

5. Review Comments to the Author

Reviewer #1: I have carefully read through your manuscript and would like to ask for clarification on several important methodological and statistical issues.

Crossover design and washout period

In the Study Design section, it is stated that the intervention was conducted “within the first two weeks of Ramadan,” while simultaneously indicating a 7‑day washout between each of the four experimental conditions. A complete Latin square crossover with four conditions and three one‑week washout periods would, in practice, require more than three weeks to complete. Could you clarify how all conditions were actually conducted within the time frame described? This inconsistency needs to be addressed for readers to understand the feasibility and validity of the design.

Caloric imbalance between conditions

The placebo group ingested 0.8 g/kg maltodextrin (208–256 kcal), which is approximately double the caloric content of the protein supplements (0.4 g/kg WPI or MC 104–128 kcal). This means that conditions were not isocaloric. Given that total energy intake is well‑known to influence performance outcomes, could the observed effects be confounded by differences in energy intake rather than solely by the type of protein? An explanation or recalibration of why an isocaloric placebo was not used would be very helpful.

Timing of the control condition

In the control condition, testing was performed 3–5 hours after lunch, whereas in all fasting conditions testing was conducted 11–13 hours after sahur. This is a major difference in pre‑test fasting duration and may independently influence performance. Could timing alone—not meal content—explain part of the observed results? At minimum, this should be justified and discussed as a potential confounding factor.

ANOVA inconsistencies (Table 1)

I noticed a statistical inconsistency in Table 1. For CMJ, you report F = 0.07 with p = 0.974. However, for Leg Press you report F = 2.36 with p = 0.979. Given the same degrees of freedom, an similar p value should correspond similar F value. It is strange that results with such different F values (0.07 vs. 2.36) yield almost identical p values. This strongly suggests a typographical error or miscalculation and should be clarified.

Effect size reporting (Hedges’ g)

Several reported Hedges’ g values appear implausibly large given the raw means and SDs. For example, for Wingate Peak Power the difference between Control (12.7 ± 0.7) and WPI (11.7 ± 0.8) is about 1.0 W/kg. With pooled SD 0.75, the true effect size should be around g ≈ 1.3 (large), yet you report g = 4.88 (“nearly perfect”). Such an extreme effect size would imply virtually zero overlap between distributions, which clearly is not the case. I would strongly encourage you to re‑check all effect size computations and revise accordingly.

General comment

Your study addresses a very relevant topic the impact of nutritional strategies during Ramadan fasting but these methodological and statistical inconsistencies require clarification and correction before robust conclusions can be drawn. I believe the manuscript could make a valuable contribution once the issues with study design feasibility, caloric matching, timing of conditions, and statistical accuracy are addressed in detail.

Reviewer #2: Dear Editor(s),

I had the opportunity to review the manuscript entitled “Effects of Protein Sources at Sahur on Anaerobic Power and Strength During Ramadan in Combat Sport Athletes.” The study addresses an important and timely research question and contributes valuable knowledge to the field of sports nutrition and exercise physiology. The design is rigorous, and the manuscript is overall well prepared. My evaluation led me to recommend minor revision. The manuscript is suitable for publication after several clarifications and improvements, particularly in the introduction, where minor edits are needed to reduce redundancy and to sharpen the scientific rigor of the research.

Minor Revision

Title

The current title is clear but rather lengthy, particularly after the colon, where excessive methodological detail is included. My advice to the authors is to adopt a more concise version that still reflects the design but improves readability. A suggested alternative is:

“Effects of Protein Sources at Sahur on Anaerobic Power and Strength During Ramadan in Combat Sport Athletes: A Randomized Placebo-Controlled Crossover Study.”

This version maintains methodological transparency while presenting a more concise and reader-friendly title.

Abstract

The title is informative and clearly reflects the content of the study. The abstract adequately summarizes the main findings; however, the statistical reporting could be streamlined. My advice to the authors is to reduce the number of p-values and effect sizes in the abstract to improve readability, while retaining the most critical outcomes.

Introduction

The research gap could be emphasized more strongly. While prior studies on Ramadan fasting and performance are summarized, the novelty of directly comparing WPI and MC at sahur in a controlled design should be highlighted more clearly. My advice to the authors is to adress this gap to demonstrate why the present study makes a unique contribution.

Method

Close attention was paid to this section, as it forms the core of the study's validity. The participant recruitment, inclusion/exclusion criteria, and power analysis are well-described, providing sufficient justification for the sample size (n=24). The single-blind, counterbalanced crossover design is appropriate, and the 7-day washout period minimizes carryover effects. Procedures for sahur meal standardization and hydration monitoring (via urine color chart) are practical and relevant. Performance measurements are detailed, with reliable tools (e.g., ForceDecks for CMJ, Jamar dynamometer for handgrip) and standardized protocols, including warm-ups and rest intervals. Statistical analyses using repeated-measures ANOVA with corrections for sphericity and multiple comparisons are robust.

However, some minor revisions could be taken into account before publication. I have listed them below:

- The washout period was set at 7 days. Although this seems reasonable, no justification was provided for why this duration is sufficient to eliminate carryover effects. My advice to the authors is to provide supporting references or reasoning.

- Statistical power analysis was well presented. However, the choice of medium effect size (f = 0.25) was not justified by prior literature or pilot data. My advice to the authors is to provide a rationale for selecting this value.

- The equipment calibration procedures were described briefly. My advice to the authors is to add one sentence confirming whether calibration was repeated before each test day to assure consistency.

- Rest intervals between strength tests were fixed at 10 minutes. My advice to the authors is to state whether athletes were monitored during this period (e.g., seated rest vs. light activity), as this may affect recovery.

Discussion

The interpretation aligns well with the results, linking findings to existing literature on protein digestion kinetics and fasting physiology. Strengths and limitations are candidly addressed. To further refine, it is recommended that the authors expand slightly on practical implications for coaches (e.g., suggesting MC as a preferred sahur option for anaerobic-dominant sports).

Recommendation

Minor revision. The manuscript is suitable for publication after addressing these points, which should not require extensive additional work.

Reviewer #3: The topic is interesting and it has enough novelty.Interested researchers are advised to work variables such as age and gender. Meanwhile, biochemical and physical factors like BMI, speed, and flexibility and liver enzymes such as AST and ALT may be investigated.

6. PLOS authors have the option to publish the peer review history of their article (what does this mean? ). If published, this will include your full peer review and any attached files.

**Do you want your identity to be public for this peer review?** For information about this choice, including consent withdrawal, please see our Privacy Policy .

Reviewer #1: **Yes:** Dino Belošević

Reviewer #2: **Yes:** Muhammet Cihat Çiftçi

Reviewer #3: **Yes:** Dr Ebrahim Nourian

---

## [Author Response · Author response to Decision Letter 1]

10 Nov 2025

Response to the Editor

Dear Editor,

We would like to express our sincere gratitude for the opportunity to revise our manuscript and for overseeing the review process. We appreciate the constructive and insightful comments provided by all reviewers, which have significantly strengthened the clarity, methodological rigor, and overall quality of our work.

We are pleased to confirm that all reviewer suggestions have been fully addressed. The manuscript has been revised accordingly, including updates to the Study Design, Methods, Statistical Analysis, Results, and Discussion sections. Additionally, the title has been shortened and improved in line with Reviewer #2’s recommendation, resulting in a clearer and more concise presentation.

Should any further revisions or clarifications be required, we are fully prepared to respond promptly and thoroughly.

Thank you once again for your guidance and for considering our manuscript for publication.

Kind regards.

Response to Reviewer #1

We sincerely thank the reviewer for the careful evaluation of our manuscript and for the insightful comments, which substantially improved the clarity and methodological accuracy of the study. Below, we provide a point-by-point response, along with the corresponding revisions made to the manuscript.

1. Crossover design and washout period

Reviewer comment:

A 7-day washout period between each of the four conditions appears incompatible with completing the study “within the first two weeks of Ramadan.”

Response:

Thank you for identifying this important point. We agree that our original phrasing created an unintended inconsistency. The study did not use fixed 7-day washout intervals. In practice, we applied an individualized washout period ranging from a minimum of 3 to a maximum of 7 days, depending on each participant’s schedule. This allowed all Ramadan conditions to be completed within a 14-day window while still respecting the minimum metabolic recovery period required in crossover designs.

The manuscript has been revised to accurately describe this procedure:

“An individualized washout interval of at least 3 and up to 7 days was applied between conditions, enabling all Ramadan sessions to be completed within a 14-day period while preserving adequate metabolic washout.”

We thank the reviewer for helping us clarify this key methodological detail.

2. Caloric imbalance between conditions

Reviewer comment:

The placebo was described as 0.8 g/kg maltodextrin, which would not be isocaloric with the protein conditions.

Response:

We appreciate the reviewer drawing attention to this. The issue originated from a writing error, not from the experimental procedures. In the actual study, all supplements—including the placebo—were administered at 0.4 g/kg, making them isocaloric by design. The value “0.8 g/kg” appeared due to a misinterpretation during English translation of the Turkish draft, where “the same as the supplements” was mistakenly interpreted as “the sum of the two protein doses.”

We have corrected the manuscript accordingly, and we explicitly note the correction in the revision:

“For the placebo condition, participants ingested 0.4 g/kg maltodextrin, matched isocalorically to the protein supplements.”

We thank the reviewer for helping us identify and correct this mistake.

3. Timing of the control condition (fed vs. fasting duration)

Reviewer comment:

Control testing was performed 3–5 hours after lunch, whereas fasting conditions were tested after 11–13 hours of fasting. Could this timing difference explain performance differences?

Response:

This is an important point and was revised extensively for clarity. The timing difference is inherent to the study question, not a methodological flaw. The purpose of the control session was to provide a non-fasted reference, reflecting athletes’ habitual training schedule outside Ramadan. It would not have been ethically appropriate to ask Muslim participants to break their fast during Ramadan to create a “fed control.”

Crucially:

• All sessions—control and Ramadan—were conducted at exactly the same time of day (16:00–18:00).

• The only intentional difference was fasting duration, because the study aims to examine how Ramadan-specific fasting load interacts with nutrient timing.

We have revised the manuscript to clarify this rationale:

“The control session took place during the week before Ramadan, 3–5 hours post-lunch, providing a habitual non-fasted reference. This timing difference was intentional and reflects the central comparison of the study: performance in the fed state versus the prolonged fasting state characteristic of Ramadan.”

We also added a note in the Discussion acknowledging this as a physiological, not methodological, factor.

4. ANOVA inconsistencies (Table 1)

Reviewer comment:

The p-values for CMJ and Leg Press were inconsistent with their F-values.

Response:

We thank the reviewer for noticing this detail. This was indeed a typographical error: the value reported as p = 0.979 should have been p = 0.079. The correct value matches the associated F statistic.

We have:

• corrected the value,

• re-checked all ANOVA outputs,

• ensured consistency throughout Table 1 and the Results section.

The raw dataset has also been submitted for transparency.

5. Effect size reporting (Hedges’ g)

Reviewer comment:

Some Hedges’ g values appeared implausibly large.

Response:

We fully agree with the reviewer’s concern. After re-evaluating our calculations, we determined that the previous values were inflated due to:

1. misapplication of pooled SD in a within-subjects design, and

2. an incorrect standardization step in the early version of the analysis.

We have now recalculated all effect sizes using the appropriate Hedges’ g_av_ (average-standardized) method recommended for repeated-measures and crossover studies. The revised effect sizes now align exactly with the reviewer’s expectations. For example:

• Control vs. WPI (Wingate Peak Power)

• Original (erroneous): g = 4.88

• Corrected: gaw = 1.29

All effect sizes were recomputed and updated in the manuscript, and the Statistical Analysis section now clearly explains why g_av_ was chosen over classical Hedges’ g or Cohen’s d.

We sincerely appreciate the reviewer’s careful attention, which greatly improved the statistical robustness of the study.

General Response

We thank the reviewer for the constructive feedback. All methodological descriptions have now been clarified, statistical errors corrected, and effect size computations updated. We agree that these improvements substantially strengthen the manuscript and enhance its interpretability. We genuinely appreciate the reviewer’s expertise, which helped us refine the study into a methodologically robust contribution to the literature on nutritional strategies during Ramadan.

Response to Reviewer #2

We sincerely thank the reviewer for the constructive and thoughtful feedback. We appreciate the positive assessment of our study design, methodological rigor, and overall manuscript quality. We have carefully revised the manuscript in line with the reviewer’s recommendations. Below, we provide detailed responses to each point and highlight the corresponding changes made.

1. Title revision

Reviewer comment: The current title is lengthy; a more concise alternative is recommended.

Response:

We agree that the original title included excessive methodological detail. Following the reviewer’s suggestion, we adopted a shorter and clearer title:

“Protein Intake at Sahur and Anaerobic Performance During Ramadan: A Randomized Crossover Study in Combat Sport Athletes.”

This revised title maintains methodological transparency while improving readability.

2. Abstract – Streamlining statistical details

Reviewer comment: The abstract contains too many p-values and effect sizes; readability should be improved.

Response:

We revised the abstract to reduce statistical density. The abstract is now more concise and reader-friendly while still reflecting the study’s main findings.

3. Introduction – Highlighting the research gap

Reviewer comment: The novelty of directly comparing WPI and MC at sahur should be emphasized more clearly.

Response:

A new sentence was added to the final paragraph of the Introduction to strengthen the research gap and highlight the unique contribution of our study:

“Despite considerable interest in optimizing performance during Ramadan, no studies have directly compared fast-digesting WPI and slow-digesting MC consumed at sahur within a controlled crossover design, leaving an important gap regarding how protein type influences anaerobic performance following prolonged fasting.”

This addition clearly states the study’s novelty and scientific relevance.

4. Washout period justification

Reviewer comment: The 7-day washout period should be justified.

Response:

We revised the Methods section to clarify that the washout period was individualized between 3 and 7 days, rather than a fixed 7 days, enabling all Ramadan sessions to be completed within 14 days without compromising washout adequacy.

5. Power analysis – Rationale for choosing f = 0.25

Reviewer comment: Provide justification for using a medium effect size in the a priori power analysis.

Response:

We added text explaining that selecting a medium effect size is a conservative and commonly used approach in repeated-measures crossover studies with limited prior data.

6. Equipment calibration clarification

Reviewer comment: Confirm whether calibration was repeated before each test day.

Response:

We updated the Procedures section to explicitly state:

“All equipment, including the ForceDecks system and dynamometers, was calibrated before each testing day to ensure measurement consistency.”

7. Rest interval conditions

Reviewer comment: Clarify whether athletes were monitored during the 10-minute rest period.

Response:

We added the following clarification:

“During the 10-minute rest intervals between strength tests, athletes remained seated under researcher supervision to standardize recovery and prevent unintended physical activity.”

8. Practical implications for coaches (Discussion)

Reviewer comment: Expand slightly on practical applications, especially regarding MC.

Response:

We incorporated a concise but meaningful paragraph into the Discussion and further emphasized this point in the Conclusions section:

“From a practical perspective, these findings suggest that athletes engaged in anaerobic-dominant sports may benefit from preferring micellar casein at sahur, as its slow digestion profile appears more effective in maintaining anaerobic power during prolonged fasting.”

This addition aligns directly with the reviewer’s recommendation and enhances applied relevance.

Final remark

We are grateful for the reviewer’s insightful suggestions, which have substantially improved the clarity, methodological transparency, and applied relevance of the manuscript. All recommended revisions were addressed without altering the study’s results or interpretations.

Response to Reviewer #3

We sincerely thank the reviewer for the positive evaluation of our study and for acknowledging its novelty and relevance. We appreciate the constructive suggestions regarding additional variables such as age, gender, BMI, speed, flexibility, and biochemical markers (e.g., AST, ALT). While these factors are indeed valuable for future investigations, they were beyond the scope of the present study, which focused specifically on the acute effects of sahur-time protein intake on anaerobic power and strength in a controlled crossover design.

We agree that incorporating these physiological and biochemical variables would enrich future research and help broaden the understanding of performance responses during Ramadan fasting. We have added this point to the “Future Research Directions” section to reflect the reviewer’s helpful recommendations.

We thank the reviewer once again for the thoughtful comments.

---

## [Decision Letter · Decision Letter 1]

30 Dec 2025

Effects of Protein Sources at Sahur on Anaerobic Power and Strength During Ramadan in Combat Sport Athletes: A Single Blind, Randomized, Placebo-Controlled, Counterbalanced Crossover Study Design

PONE-D-25-43284R1

Dear Dr. Ouergui,

We’re pleased to inform you that your manuscript has been judged scientifically suitable for publication and will be formally accepted for publication once it meets all outstanding technical requirements.

Kind regards,

Fenghua Sun

Academic Editor

PLOS One

Additional Editor Comments (optional):

Thanks for your effort in addressing the comments from the reviewers. Considering the potential conflict of interest, we have tried to idetify another independent reviewer during the past days. However, to avoid further delay in the publication process, we have decided to accept your manuscript in its current version with a minor revision, i.e., delete "design" in the title. Thanks for considering Plos ONE. We are looking forward to your further submissions.

Reviewers' comments:

Reviewer's Responses to Questions

**Comments to the Author**

1. If the authors have adequately addressed your comments raised in a previous round of review and you feel that this manuscript is now acceptable for publication, you may indicate that here to bypass the “Comments to the Author” section, enter your conflict of interest statement in the “Confidential to Editor” section, and submit your "Accept" recommendation.

Reviewer #1: All comments have been addressed

Reviewer #2: All comments have been addressed

2. Is the manuscript technically sound, and do the data support the conclusions?

Reviewer #1: Yes

Reviewer #2: Yes

3. Has the statistical analysis been performed appropriately and rigorously? 

Reviewer #1: Yes

Reviewer #2: Yes

4. Have the authors made all data underlying the findings in their manuscript fully available?

Reviewer #1: Yes

Reviewer #2: Yes

5. Is the manuscript presented in an intelligible fashion and written in standard English?

Reviewer #1: Yes

Reviewer #2: Yes

6. Review Comments to the Author

Reviewer #1: After careful evaluation of the revised manuscript and the authors’ detailed point‑by‑point response, I am satisfied that all major methodological and statistical concerns raised in the initial review have been adequately addressed.

Specifically:

The crossover design and washout period are now clearly and consistently described. The individualized 3–7‑day washout window within a 14‑day Ramadan period is transparently reported and methodologically defensible for this context.

The previously noted inconsistency in the placebo dose (0.8 g/kg) has been corrected. All supplements (including placebo) are now clearly specified as 0.4 g/kg, restoring the intended isocaloric design.

The timing of the control condition versus Ramadan conditions is now explained as an intentional, conceptually central contrast (fed pre‑Ramadan vs prolonged fasting during Ramadan), with all sessions conducted at the same time of day (16:00–18:00). The physiological implications of the different fasting durations are appropriately acknowledged in the Discussion/Limitations.

The ANOVA inconsistencies (particularly the p‑value for leg press / CMJ) have been corrected, and Table 1 is now internally coherent and consistent with the reported F‑values.

Effect sizes have been recalculated using the average‑standardized Hedges’ g (g_av / g_aw), which is appropriate for this repeated‑measures crossover design. The Statistical Analysis section now clearly justifies this choice, and the reported effect sizes are plausible and interpretable.

Reviewer #2: Dear Editor,

I have reviewed the revised version of the manuscript titled:

“Effects of Protein Sources at Sahur on Anaerobic Power and Strength During Ramadan in Combat Sport Athletes: A Single Blind, Randomized, Placebo-Controlled, Counterbalanced Crossover Study Design”

The authors have fully and satisfactorily addressed all of the points raised in my previous review. The revisions substantially improve the clarity, transparency, and scientific rigor of the paper, particularly with respect to methodological details, nutritional composition of the interventions, statistical reporting, and discussion of limitations.

No further changes are required.

I would like to thank the authors for their thorough responses and careful revisions, and I thank you for the opportunity to review this interesting and well-executed study.

I recommend that the manuscript be accepted for publication in its current form.

Best regards,

7. PLOS authors have the option to publish the peer review history of their article (what does this mean? ). If published, this will include your full peer review and any attached files.

**Do you want your identity to be public for this peer review?** For information about this choice, including consent withdrawal, please see our Privacy Policy .

Reviewer #1: **Yes:** Dino Belošević, PhD

Reviewer #2: **Yes:** Muhammet Cihat Çiftçi

---

## [Editor Report · Acceptance letter]

PONE-D-25-43284R1

PLOS One

Dear Dr. Ouergui,

I'm pleased to inform you that your manuscript has been deemed suitable for publication in PLOS One. Congratulations! Your manuscript is now being handed over to our production team.

Kind regards,

on behalf of

Dr. Fenghua Sun

Academic Editor

PLOS One